# A Simple Baseline for Bayesian Uncertainty in Deep Learning

**Wesley J. Maddox**[*1]    **Timur Garipov**[*2]    **Pavel Izmailov**[*1]
**Dmitry Vetrov**[2,3]    **Andrew Gordon Wilson**[1]

[1] New York University
[2] Samsung AI Center Moscow
[3] Samsung-HSE Laboratory, National Research University Higher School of Economics

## Abstract

We propose SWA-Gaussian (SWAG), a simple, scalable, and general purpose approach for uncertainty representation and calibration in deep learning. Stochastic Weight Averaging (SWA), which computes the first moment of stochastic gradient descent (SGD) iterates with a modified learning rate schedule, has recently been shown to improve generalization in deep learning. With SWAG, we fit a Gaussian using the SWA solution as the first moment and a low rank plus diagonal covariance also derived from the SGD iterates, forming an approximate posterior distribution over neural network weights; we then sample from this Gaussian distribution to perform Bayesian model averaging. We empirically find that SWAG approximates the shape of the true posterior, in accordance with results describing the stationary distribution of SGD iterates. Moreover, we demonstrate that SWAG performs well on a wide variety of tasks, including out of sample detection, calibration, and transfer learning, in comparison to many popular alternatives including MC dropout, KFAC Laplace, SGLD, and temperature scaling.

## 1 Introduction

Ultimately, machine learning models are used to make decisions. Representing uncertainty is crucial for decision making. For example, in medical diagnoses and autonomous vehicles we want to protect against rare but costly mistakes. Deep learning models typically lack a representation of uncertainty, and provide overconfident and miscalibrated predictions [e.g., 21, 12].

Bayesian methods provide a natural probabilistic representation of uncertainty in deep learning [e.g., 3, 24, 5], and previously had been a gold standard for inference with neural networks [38]. However, existing approaches are often highly sensitive to hyperparameter choices, and hard to scale to modern datasets and architectures, which limits their general applicability in modern deep learning.

In this paper we propose a different approach to Bayesian deep learning: we use the information contained in the SGD trajectory to efficiently approximate the posterior distribution over the weights of the neural network. We find that the Gaussian distribution fitted to the first two moments of SGD iterates, with a modified learning rate schedule, captures the local geometry of the posterior surprisingly well. Using this Gaussian distribution we are able to obtain convenient, efficient, accurate and well-calibrated predictions in a broad range of tasks in computer vision. In particular, our contributions are the following:

- In this work we propose SWAG (SWA-Gaussian), a scalable approximate Bayesian inference technique for deep learning. SWAG builds on Stochastic Weight Averaging [20], which

---

[*]Equal contribution. Correspondence to wjm363 AT nyu.edu

computes an average of SGD iterates with a high constant learning rate schedule, to provide improved generalization in deep learning and the interpretation of SGD as approximate Bayesian inference [34]. SWAG additionally computes a low-rank plus diagonal approximation to the covariance of the iterates, which is used together with the SWA mean, to define a Gaussian posterior approximation over neural network weights.

- SWAG is motivated by the theoretical analysis of the stationary distribution of SGD iterates [e.g., 34, 6], which suggests that the SGD trajectory contains useful information about the geometry of the posterior. In Appendix 2 we show that the assumptions of Mandt et al. [34] do not hold for deep neural networks, due to non-convexity and over-parameterization (with further analysis in the supplementary material). However, we find in Section 4 that in the low-dimensional subspace spanned by SGD iterates the shape of the posterior distribution is approximately Gaussian within a basin of attraction. Further, SWAG is able to capture the geometry of this posterior remarkably well.

- In an exhaustive empirical evaluation we show that SWAG can provide well-calibrated uncertainty estimates for neural networks across many settings in computer vision. In particular SWAG achieves higher test likelihood compared to many state-of-the-art approaches, including MC-Dropout [9], temperature scaling [12], SGLD [46], KFAC-Laplace [43] and SWA [20] on CIFAR-10, CIFAR-100 and ImageNet, on a range of architectures. We also demonstrate the effectiveness of SWAG for out-of-domain detection, and transfer learning. While we primarily focus on image classification, we show that SWAG can significantly improve test perplexities of LSTM networks on language modeling problems, and in Appendix 7 we also compare SWAG with Probabilistic Back-propagation (PBP) [16], Deterministic Variational Inference (DVI) [47], and Deep Gaussian Processes [4] on regression problems.

- We release PyTorch code at `https://github.com/wjmaddox/swa_gaussian`.

## 2 Related Work

### 2.1 Bayesian Methods

Bayesian approaches represent uncertainty by placing a distribution over model parameters, and then marginalizing these parameters to form a whole predictive distribution, in a procedure known as Bayesian model averaging. In the late 1990s, Bayesian methods were the state-of-the-art approach to learning with neural networks, through the seminal works of Neal [38] and MacKay [32]. However, modern neural networks often contain millions of parameters, the posterior over these parameters (and thus the loss surface) is highly non-convex, and mini-batch approaches are often needed to move to a space of good solutions [22]. For these reasons, Bayesian approaches have largely been intractable for modern neural networks. Here, we review several modern approaches to Bayesian deep learning.

**Markov chain Monte Carlo (MCMC)** was at one time a gold standard for inference with neural networks, through the Hamiltonian Monte Carlo (HMC) work of Neal [38]. However, HMC requires full gradients, which is computationally intractable for modern neural networks. To extend the HMC framework, stochastic gradient HMC (SGHMC) was introduced by Chen et al. [5] and allows for stochastic gradients to be used in Bayesian inference, crucial for both scalability and exploring a space of solutions that provide good generalization. Alternatively, stochastic gradient Langevin dynamics (SGLD) [46] uses first order Langevin dynamics in the stochastic gradient setting. Theoretically, both SGHMC and SGLD asymptotically sample from the posterior in the limit of infinitely small step sizes. In practice, using finite learning rates introduces approximation errors (see e.g. [34]), and tuning stochastic gradient MCMC methods can be quite difficult.

**Variational Inference:** Graves [11] suggested fitting a Gaussian variational posterior approximation over the weights of neural networks. This technique was generalized by Kingma and Welling [26] which proposed the *reparameterization trick* for training deep latent variable models; multiple variational inference methods based on the reparameterization trick were proposed for DNNs [e.g., 25, 3, 36, 31]. While variational methods achieve strong performance for moderately sized networks, they are empirically noted to be difficult to train on larger architectures such as deep residual networks [15]; Blier and Ollivier [2] argue that the difficulty of training is explained by variational methods

providing insufficient data compression for DNNs despite being designed for data compression (minimum description length). Recent key advances [31, 47] in variational inference for deep learning typically focus on smaller-scale datasets and architectures. An alternative line of work re-interprets noisy versions of optimization algorithms: for example, noisy Adam [23] and noisy KFAC [50], as approximate variational inference.

**Dropout Variational Inference:**   Gal and Ghahramani [9] used a spike and slab variational distribution to view dropout at test time as approximate variational Bayesian inference. Concrete dropout [10] extends this idea to optimize the dropout probabilities as well. From a practical perspective, these approaches are quite appealing as they only require ensembling dropout predictions at test time, and they were succesfully applied to several downstream tasks [21, 37].

**Laplace Approximations**   assume a Gaussian posterior, $\mathcal{N}(\theta^*, \mathcal{I}(\theta^*)^{-1})$, where $\theta^*$ is a MAP estimate and $\mathcal{I}(\theta^*)^{-1}$ is the inverse of the Fisher information matrix (expected value of the Hessian evaluated at $\theta^*$). It was notably used for Bayesian neural networks in MacKay [33], where a diagonal approximation to the inverse of the Hessian was utilized for computational reasons. More recently, Kirkpatrick et al. [27] proposed using diagonal Laplace approximations to overcome catastrophic forgetting in deep learning. Ritter et al. [43] proposed the use of either a diagonal or block Kronecker factored (KFAC) approximation to the Hessian matrix for Laplace approximations, and Ritter et al. [42] successfully applied the KFAC approach to online learning scenarios.

## 2.2   SGD Based Approximations

Mandt et al. [34] proposed to use the iterates of averaged SGD as an MCMC sampler, after analyzing the dynamics of SGD using tools from stochastic calculus. From a frequentist perspective, Chen et al. [6] showed that under certain conditions a batch means estimator of the sample covariance matrix of the SGD iterates converges to $A = \mathcal{H}(\theta)^{-1}C(\theta)\mathcal{H}(\theta)^{-1}$, where $\mathcal{H}(\theta)^{-1}$ is the inverse of the Hessian of the log likelihood and $C(\theta) = \mathbb{E}(\nabla \log p(\theta)\nabla \log p(\theta)^T)$ is the covariance of the gradients of the log likelihood. Chen et al. [6] then show that using $A$ and the sample average of the iterates for a Gaussian approximation produces well calibrated confidence intervals of the parameters and that the variance of these estimators achieves the Cramer Rao lower bound (the minimum possible variance). A description of the asymptotic covariance of the SGD iterates dates back to Ruppert [44] and Polyak and Juditsky [41], who show asymptotic convergence of Polyak-Ruppert averaging.

## 2.3   Methods for Calibration of DNNs

Lakshminarayanan et al. [29] proposed using ensembles of several networks for enhanced calibration, and incorporated an adversarial loss function to be used when possible as well. Outside of probabilistic neural networks, Guo et al. [12] proposed temperature scaling, a procedure which uses a validation set and a single hyperparameter to rescale the logits of DNN outputs for enhanced calibration. Kuleshov et al. [28] propose calibrated regression using a similar rescaling technique.

# 3   SWA-Gaussian for Bayesian Deep Learning

In this section we propose SWA-Gaussian (SWAG) for Bayesian model averaging and uncertainty estimation. In Section 3.2, we review stochastic weight averaging (SWA) [20], which we view as estimating the mean of the stationary distribution of SGD iterates. We then propose SWA-Gaussian in Sections 3.3 and 3.4 to estimate the covariance of the stationary distribution, forming a Gaussian approximation to the posterior over weight parameters. With SWAG, uncertainty in weight space is captured with minimal modifications to the SWA training procedure. We then present further theoretical and empirical analysis for SWAG in Section 4.

## 3.1   Stochastic Gradient Descent (SGD)

Standard training of deep neural networks (DNNs) proceeds by applying stochastic gradient descent on the model weights $\theta$ with the following update rule:

$$\Delta\theta_t = -\eta_t \left( \frac{1}{B}\sum_{i=1}^{B} \nabla_\theta \log p(y_i|f_\theta(x_i)) - \frac{\nabla_\theta \log p(\theta)}{N} \right),$$

where the learning rate is $\eta$, the $i$th input (e.g. image) and label are $\{x_i, y_i\}$, the size of the whole training set is $N$, the size of the batch is $B$, and the DNN, $f$, has weight parameters $\theta$.[2] The loss function is a negative log likelihood $-\sum_i \log p(y_i | f_\theta(x_i))$, combined with a regularizer $\log p(\theta)$. This type of maximum likelihood training does not represent uncertainty in the predictions or parameters $\theta$.

## 3.2 Stochastic Weight Averaging (SWA)

The main idea of SWA [20] is to run SGD with a constant learning rate schedule starting from a pre-trained solution, and to average the weights of the models it traverses. Denoting the weights of the network obtained after epoch $i$ of SWA training $\theta_i$, the SWA solution after $T$ epochs is given by $\theta_{\text{SWA}} = \frac{1}{T} \sum_{i=1}^{T} \theta_i$. A high constant learning rate schedule ensures that SGD explores the set of possible solutions instead of simply converging to a single point in the weight space. Izmailov et al. [20] argue that conventional SGD training converges to the boundary of the set of high-performing solutions; SWA on the other hand is able to find a more centered solution that is robust to the shift between train and test distributions, leading to improved generalization performance. SWA and related ideas have been successfully applied to a wide range of applications [see e.g. 1, 48, 49, 40]. A related but different procedure is Polyak-Ruppert averaging [41, 44] in stochastic convex optimization, which uses a learning rate decaying to zero. Mandt et al. [34] interpret Polyak-Ruppert averaging as a sampling procedure, with convergence occurring to the true posterior under certain strong conditions. Additionally, they explore the theoretical feasibility of SGD (and averaged SGD) as an approximate Bayesian inference scheme; we test their assumptions in Appendix 1.

## 3.3 SWAG-Diagonal

We first consider a simple diagonal format for the covariance matrix. In order to fit a diagonal covariance approximation, we maintain a running average of the second uncentered moment for each weight, and then compute the covariance using the following standard identity at the end of training: $\overline{\theta^2} = \frac{1}{T} \sum_{i=1}^{T} \theta_i^2$, $\Sigma_{\text{diag}} = \text{diag}(\overline{\theta^2} - \theta_{\text{SWA}}^2)$; here the squares in $\theta_{\text{SWA}}^2$ and $\theta_i^2$ are applied elementwise. The resulting approximate posterior distribution is then $\mathcal{N}(\theta_{\text{SWA}}, \Sigma_{\text{Diag}})$. In our experiments, we term this method SWAG-Diagonal.

Constructing the SWAG-Diagonal posterior approximation requires storing two additional copies of DNN weights: $\theta_{\text{SWA}}$ and $\overline{\theta^2}$. Note that these models do not have to be stored on the GPU. The additional computational complexity of constructing SWAG-Diagonal compared to standard training is negligible, as it only requires updating the running averages of weights once per epoch.

## 3.4 SWAG: Low Rank plus Diagonal Covariance Structure

We now describe the full SWAG algorithm. While the diagonal covariance approximation is standard in Bayesian deep learning [3, 27], it can be too restrictive. We extend the idea of diagonal covariance approximations to utilize a more flexible low-rank plus diagonal posterior approximation. SWAG approximates the sample covariance $\Sigma$ of the SGD iterates along with the mean $\theta_{\text{SWA}}$.[3]

Note that the sample covariance matrix of the SGD iterates can be written as the sum of outer products, $\Sigma = \frac{1}{T-1} \sum_{i=1}^{T} (\theta_i - \theta_{\text{SWA}})(\theta_i - \theta_{\text{SWA}})^\top$, and is of rank $T$. As we do not have access to the value of $\theta_{\text{SWA}}$ during training, we approximate the sample covariance with $\Sigma \approx \frac{1}{T-1} \sum_{i=1}^{T} (\theta_i - \bar{\theta}_i)(\theta_i - \bar{\theta}_i)^\top = \frac{1}{T-1} DD^\top$, where $D$ is the deviation matrix comprised of columns $D_i = (\theta_i - \bar{\theta}_i)$, and $\bar{\theta}_i$ is the running estimate of the parameters' mean obtained from the first $i$ samples. To limit the rank of the estimated covariance matrix we only use the last $K$ of $D_i$ vectors corresponding to the last $K$

epochs of training. Here $K$ is the rank of the resulting approximation and is a hyperparameter of the method. We define $\widehat{D}$ to be the matrix with columns equal to $D_i$ for $i = T - K + 1, \ldots, T$.

We then combine the resulting low-rank approximation $\Sigma_{\text{low-rank}} = \frac{1}{K-1} \cdot \widehat{D}\widehat{D}^\top$ with the diagonal approximation $\Sigma_{\text{diag}}$ of Section 3.3. The resulting approximate posterior distribution is a Gaussian with the SWA mean $\theta_{\text{SWA}}$ and summed covariance: $\mathcal{N}(\theta_{\text{SWA}}, \frac{1}{2} \cdot (\Sigma_{\text{diag}} + \Sigma_{\text{low-rank}}))$.[4] In our experiments, we term this method SWAG. Computing this approximate posterior distribution requires storing $K$ vectors $D_i$ of the same size as the model as well as the vectors $\theta_{\text{SWA}}$ and $\overline{\theta^2}$. These models do not have to be stored on a GPU.

To sample from SWAG we use the following identity

$$\widetilde{\theta} = \theta_{\text{SWA}} + \frac{1}{\sqrt{2}} \cdot \Sigma_{\text{diag}}^{\frac{1}{2}} z_1 + \frac{1}{\sqrt{2(K-1)}}\widehat{D}z_2, \quad \text{where } z_1 \sim \mathcal{N}(0, I_d), \; z_2 \sim \mathcal{N}(0, I_K). \quad (1)$$

Here $d$ is the number of parameters in the network. Note that $\Sigma_{\text{diag}}$ is diagonal, and the product $\Sigma_{\text{diag}}^{\frac{1}{2}} z_1$ can be computed in $\mathcal{O}(d)$ time. The product $\widehat{D}z_2$ can be computed in $\mathcal{O}(Kd)$ time.

Related methods for estimating the covariance of SGD iterates were considered in Mandt et al. [34] and Chen et al. [6], but store full-rank covariance $\Sigma$ and thus scale quadratically in the number of parameters, which is prohibitively expensive for deep learning applications. We additionally note that using the deviation matrix for online covariance matrix estimation comes from viewing the online updates used in Dasgupta and Hsu [8] in matrix fashion.

The full Bayesian model averaging procedure is given in Algorithm 1. As in Izmailov et al. [20] (SWA) we update the batch normalization statistics after sampling weights for models that use batch normalization [18]; we investigate the necessity of this update in Appendix 4.4.

---

**Algorithm 1** Bayesian Model Averaging with SWAG

---

$\theta_0$: pretrained weights; $\eta$: learning rate; $T$: number of steps; $c$: moment update frequency; $K$: maximum number of columns in deviation matrix; $S$: number of samples in Bayesian model averaging

**Train** SWAG
$\overline{\theta} \leftarrow \theta_0, \; \overline{\theta^2} \leftarrow \theta_0^2$        {Initialize moments}
**for** $i \leftarrow 1, 2, ..., T$ **do**
    $\theta_i \leftarrow \theta_{i-1} - \eta\nabla_\theta \mathcal{L}(\theta_{i-1})$ {Perform SGD update}
    **if** $\texttt{MOD}(i, c) = 0$ **then**
        $n \leftarrow i/c$                {Number of models}
        $\overline{\theta} \leftarrow \dfrac{n\overline{\theta} + \theta_i}{n+1}, \; \overline{\theta^2} \leftarrow \dfrac{n\overline{\theta^2} + \theta_i^2}{n+1}$ {Moments}
        **if** $\texttt{NUM\_COLS}(\widehat{D}) = K$ **then**
            $\texttt{REMOVE\_COL}(\widehat{D}[:, 1])$
        $\texttt{APPEND\_COL}(\widehat{D}, \theta_i - \overline{\theta})$   {Store deviation}
**return** $\theta_{\text{SWA}} = \overline{\theta}, \; \Sigma_{\text{diag}} = \overline{\theta^2} - \overline{\theta}^2, \; \widehat{D}$

**Test** Bayesian Model Averaging
**for** $i \leftarrow 1, 2, ..., S$ **do**
    Draw $\widetilde{\theta}_i \sim \mathcal{N}\left(\theta_{\text{SWA}}, \frac{1}{2}\Sigma_{\text{diag}} + \frac{\widehat{D}\widehat{D}^\top}{2(K-1)}\right)$ (1)
    Update batch norm statistics with new sample.
    $p(y^*|\text{Data}) \mathrel{+}= \frac{1}{S}p(y^*|\widetilde{\theta}_i)$
**return** $p(y^*|\text{Data})$

---

## 3.5 Bayesian Model Averaging with SWAG

Maximum a-posteriori (MAP) optimization is a procedure whereby one maximizes the (log) posterior with respect to parameters $\theta$: $\log p(\theta|\mathcal{D}) = \log p(\mathcal{D}|\theta) + \log p(\theta)$. Here, the prior $p(\theta)$ is viewed as a regularizer in optimization. However, MAP is *not* Bayesian inference, since one only considers a single setting of the parameters $\hat{\theta}_{\text{MAP}} = \text{argmax}_\theta p(\theta|\mathcal{D})$ in making predictions, forming $p(y_*|\hat{\theta}_{\text{MAP}}, x_*)$, where $x_*$ and $y_*$ are test inputs and outputs.

A Bayesian procedure instead *marginalizes* the posterior distribution over $\theta$, in a Bayesian model average, for the unconditional predictive distribution: $p(y_*|\mathcal{D}, x_*) = \int p(y_*|\theta, x_*)p(\theta|\mathcal{D})d\theta$. In practice, this integral is computed through a Monte Carlo sampling procedure:
$p(y_*|\mathcal{D}, x_*) \approx \frac{1}{T}\sum_{t=1}^{T} p(y_*|\theta_t, x_*), \quad \theta_t \sim p(\theta|\mathcal{D})$.

We emphasize that in this paper we are approximating *fully Bayesian inference*, rather than MAP optimization. We develop a Gaussian approximation to the posterior from SGD iterates, $p(\theta|\mathcal{D}) \approx$

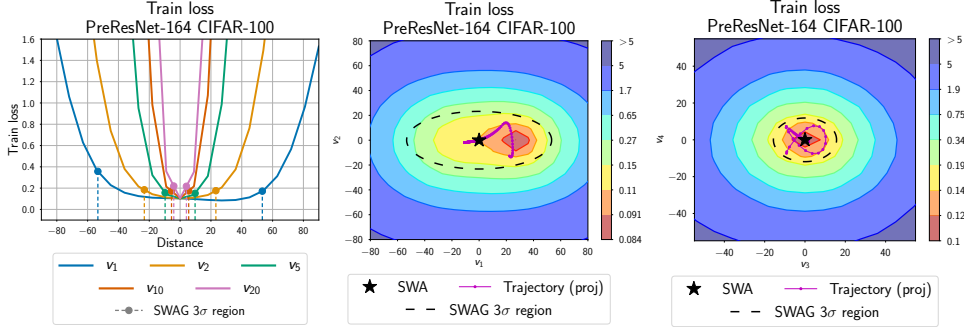

Figure 1: **Left:** Posterior joint density cross-sections along the rays corresponding to different eigenvectors of SWAG covariance matrix. **Middle:** Posterior joint density surface in the plane spanned by eigenvectors of SWAG covariance matrix corresponding to the first and second largest eigenvalues and (**Right:**) the third and fourth largest eigenvalues. All plots are produced using PreResNet-164 on CIFAR-100. The SWAG distribution projected onto these directions fits the geometry of the posterior density remarkably well.

$\mathcal{N}(\theta; \mu, \Sigma)$, and then sample from this posterior distribution to perform a Bayesian model average. In our procedure, optimization with different regularizers, to characterize the Gaussian posterior approximation, corresponds to approximate Bayesian inference with different priors $p(\theta)$.

**Prior Choice** Typically, weight decay is used to regularize DNNs, corresponding to explicit L2 regularization when SGD without momentum is used to train the model. When SGD is used *with* momentum, as is typically the case, implicit regularization still occurs, producing a vague prior on the weights of the DNN in our procedure. This regularizer can be given an explicit Gaussian-like form (see Proposition 3 of Loshchilov and Hutter [30]), corresponding to a prior distribution on the weights.

Thus, SWAG is an approximate Bayesian inference algorithm in our experiments (see Section 5) and can be applied to most DNNs without any modifications of the training procedure (as long as SGD is used with weight decay or explicit L2 regularization). Alternative regularization techniques could also be used, producing different priors on the weights. It may also be possible to similarly utilize Adam and other stochastic first-order methods, which view as a promising direction for future work.

## 4 Does the SGD Trajectory Capture Loss Geometry?

To analyze the quality of the SWAG approximation, we study the posterior density along the directions corresponding to the eigenvectors of the SWAG covariance matrix for PreResNet-164 on CIFAR-100. In order to find these eigenvectors we use randomized SVD [14].[5] In the left panel of Figure 1 we visualize the $\ell_2$-regularized cross-entropy loss $L(\cdot)$ (equivalent to the joint density of the weights and the loss with a Gaussian prior) as a function of distance $t$ from the SWA solution $\theta_{\text{SWA}}$ along the $i$-th eigenvector $v_i$ of the SWAG covariance: $\phi(t) = L(\theta_{\text{SWA}} + t \cdot \frac{v_i}{\|v_i\|})$. Figure 1 (left) shows a clear correlation between the variance of the SWAG approximation and the width of the posterior along the directions $v_i$. The SGD iterates indeed contain useful information about the shape of the posterior distribution, and SWAG is able to capture this information. We repeated the same experiment for SWAG-Diagonal, finding that there was almost no variance in these eigen-directions. Next, in Figure 1 (middle) we plot the posterior density surface in the 2-dimensional plane in the weight space spanning the two top eigenvectors $v_1$ and $v_2$ of the SWAG covariance: $\psi(t_1, t_2) = L(\theta_{\text{SWA}} + t_1 \cdot \frac{v_1}{\|v_1\|} + t_2 \cdot \frac{v_2}{\|v_2\|})$. Again, SWAG is able to capture the geometry of the posterior. The contours of constant posterior density appear remarkably well aligned with the eigenvalues of the SWAG covariance. We also present the analogous plot for the third and fourth top eigenvectors in Figure 1 (right). In Appendix 3, we additionally present similar results for PreResNet-164 on CIFAR-10 and VGG-16 on CIFAR-100.

As we can see, SWAG is able to capture the geometry of the posterior in the subspace spanned by SGD iterates. However, the dimensionality of this subspace is very low compared to the dimensionality of

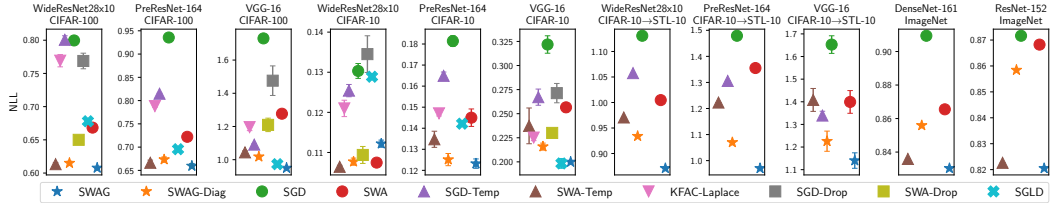

Figure 2: Negative log likelihoods for SWAG and baselines. Mean and standard deviation (shown with error-bars) over 3 runs are reported for each experiment on CIFAR datasets. SWAG (blue star) consistently outperforms alternatives, with lower negative log likelihood, with the largest improvements on transfer learning. Temperature scaling applied on top of SWA (SWA-Temp) often performs close to as well on the non-transfer learning tasks, but requires a validation set.

the weight space, and we can not guarantee that SWAG variance estimates are adequate along all directions in weight space. In particular, we would expect SWAG to under-estimate the variances along random directions, as the SGD trajectory is in a low-dimensional subspace of the weight space, and a random vector has a close-to-zero projection on this subspace with high probability. In Appendix 1 we visualize the trajectory of SGD applied to a quadratic function, and further discuss the relation between the geometry of objective and SGD trajectory. In Appendices 1 and 2, we also empirically test the assumptions behind theory relating the SGD stationary distribution to the true posterior for neural networks.

# 5 Experiments

We conduct a thorough empirical evaluation of SWAG, comparing to a range of high performing baselines, including MC dropout [9], temperature scaling [12], SGLD [46], Laplace approximations [43], deep ensembles [29], and ensembles of SGD iterates that were used to construct the SWAG approximation. In Section 5.1 we evaluate SWAG predictions and uncertainty estimates on image classification tasks. We also evaluate SWAG for transfer learning and out-of-domain data detection. We investigate the effect of hyperparameter choices and practical limitations in SWAG, such as the effect of learning rate on the scale of uncertainty, in Appendix 4.

## 5.1 Calibration and Uncertainty Estimation on Image Classification Tasks

In this section we evaluate the quality of uncertainty estimates as well as predictive accuracy for SWAG and SWAG-Diagonal on CIFAR-10, CIFAR-100 and ImageNet ILSVRC-2012 [45].

For all methods we analyze test negative log-likelihood, which reflects both the accuracy and the quality of predictive uncertainty. Following Guo et al. [12] we also consider a variant of *reliability diagrams* to evaluate the calibration of uncertainty estimates (see Figure 3) and to show the difference between a method's confidence in its predictions and its accuracy. To produce this plot for a given method we split the test data into 20 bins uniformly based on the confidence of a method (maximum predicted probability). We then evaluate the accuracy and mean confidence of the method on the images from each bin, and plot the difference between confidence and accuracy. For a well-calibrated model, this difference should be close to zero for each bin. We found that this procedure gives a more effective visualization of the actual confidence distribution of DNN predictions than the standard reliability diagrams used in Guo et al. [12] and Niculescu-Mizil and Caruana [39].

We provide tables containing the test accuracy, negative log likelihood and expected calibration error for all methods and datasets in Appendix 5.3.

**CIFAR datasets** On CIFAR datasets we run experiments with VGG-16, PreResNet-164 and WideResNet-28x10 networks. In order to compare SWAG with existing alternatives we report the results for standard SGD and SWA [20] solutions (single models), MC-Dropout [9], temperature scaling [12] applied to SWA and SGD solutions, SGLD [46], and K-FAC Laplace [43] methods. For all the methods we use our implementations in PyTorch (see Appendix 8). We train all networks for 300 epochs, starting to collect models for SWA and SWAG approximations once per epoch after epoch 160. For SWAG, K-FAC Laplace, and Dropout we use 30 samples at test time.

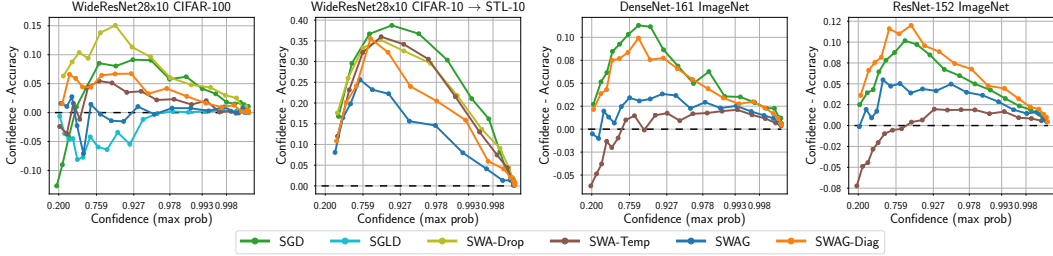

Figure 3: Reliability diagrams for WideResNet28x10 on CIFAR-100 and transfer task; ResNet-152 and DenseNet-161 on ImageNet. Confidence is the value of the max softmax output. A perfectly calibrated network has no difference between confidence and accuracy, represented by a dashed black line. Points below this line correspond to under-confident predictions, whereas points above the line are overconfident predictions. SWAG is able to substantially improve calibration over standard training (SGD), as well as SWA. Additionally, SWAG significantly outperforms temperature scaling for transfer learning (CIFAR-10 to STL), where the target data are not from the same distribution as the training data.

**ImageNet**  On ImageNet we report our results for SWAG, SWAG-Diagonal, SWA and SGD. We run experiments with DenseNet-161 [17] and Resnet-152 [15]. For each model we start from a pre-trained model available in the `torchvision` package, and run SGD with a constant learning rate for 10 epochs. We collect models for the SWAG versions and SWA 4 times per epoch. For SWAG we use 30 samples from the posterior over network weights at test-time, and use randomly sampled 10% of the training data to update batch-normalization statistics for each of the samples. For SGD with temperature scaling, we use the results reported in Guo et al. [12].

**Transfer from CIFAR-10 to STL-10**  We use the models trained on CIFAR-10 and evaluate them on STL-10 [7]. STL-10 has a similar set of classes as CIFAR-10, but the image distribution is different, so adapting the model from CIFAR-10 to STL-10 is a commonly used transfer learning benchmark. We provide further details on the architectures and hyperparameters in Appendix 8.

**Results**  We visualize the negative log-likelihood for all methods and datasets in Figure 2. On all considered tasks SWAG and SWAG diagonal perform comparably or better than all the considered alternatives, SWAG being best overall. We note that the combination of SWA and temperature scaling presents a competitive baseline. However, unlike SWAG it requires using a validation set to tune the temperature; further, temperature scaling is not effective when the test data distribution differs from train, as we observe in experiments on transfer learning from CIFAR-10 to STL-10.

Next, we analyze the calibration of uncertainty estimates provided by different methods. In Figure 3 we present reliability plots for WideResNet on CIFAR-100, DenseNet-161 and ResNet-152 on ImageNet. The reliability diagrams for all other datasets and architectures are presented in the Appendix 5.1. As we can see, SWAG and SWAG-Diagonal both achieve good calibration across the board. The low-rank plus diagonal version of SWAG is generally better calibrated than SWAG-Diagonal. We also present the expected calibration error for each of the methods, architectures and datasets in Tables A.2,3. Finally, in Tables A.8,9 we present the predictive accuracy for all of the methods, where SWAG is comparable with SWA and generally outperforms the other approaches.

## 5.2 Comparison to ensembling SGD solutions

We evaluated ensembles of independently trained SGD solutions (Deep Ensembles, [29]) on PreResNet-164 on CIFAR-100. We found that an ensemble of 3 SGD solutions has high accuracy (82.1%), but only achieves NLL 0.6922, which is *worse than a single SWAG solution* (0.6595 NLL). While the accuracy of this ensemble is high, SWAG solutions are much better calibrated. An ensemble of 5 SGD solutions achieves NLL 0.6478, which is *competitive with a single SWAG solution, that requires $5\times$ less computation to train*. Moreover, we can similarly ensemble independently trained SWAG models; an ensemble of 3 SWAG models achieves NLL of 0.6178.

We also evaluated ensembles of SGD iterates that were used to construct the SWAG approximation (SGD-Ens) for all of our CIFAR models. SWAG has higher NLL than SGD-Ens on VGG-16, but

much lower NLL on the larger PreResNet-164 and WideResNet28x10; the results for accuracy and ECE are analogous.

### 5.3 Out-of-Domain Image Detection

To evaluate SWAG on out-of-domain data detection we train a WideResNet as described in section 5.1 on the data from five classes of the CIFAR-10 dataset, and then analyze predictions of SWAG variants along with the baselines on the full test set. We expect the outputted class probabilities on objects that belong to classes that were not present in the training data to have high-entropy reflecting the model's high uncertainty in its predictions, and considerably lower entropy on the images that are similar to those on which the network was trained. We plot the histograms of predictive entropies on the in-domain and out-of-domain in Figure A.A7 for a qualitative comparison and report the symmetrized KL divergence between the binned in and out of sample distributions in Table 1, finding that SWAG and Dropout perform best on this measure. Additional details are in Appendix 5.2.

### 5.4 Language Modeling with LSTMs

We next apply SWAG to an LSTM network on language modeling tasks on Penn Treebank and WikiText-2 datasets. In Appendix 6 we demonstrate that SWAG easily outperforms both SWA and NT-ASGD [35], a strong baseline for LSTM training, in terms of test and validation perplexities.

We compare SWAG to SWA and the NT-ASGD method [35], which is a strong baseline for training LSTM models. The main difference between SWA and NT-ASGD, which is also based on weight averaging, is that NT-ASGD starts weight averaging much earlier than SWA: NT-ASGD switches to ASGD (averaged SGD) typically around epoch 100 while with SWA we start averaging after pre-training for 500 epochs. We report test and validation perplexities for different methods and datasets in Table 1.

As we can see, SWA substantially improves perplexities on both datasets over NT-ASGD. Further, we observe that SWAG is able to substantially improve test perplexities over the SWA solution.

Table 1: Validation and Test perplexities for NT-ASGD, SWA and SWAG on Penn Treebank and WikiText-2 datasets.

| Method | PTB val | PTB test | WikiText-2 val | WikiText-2 test |
|--------|---------|----------|----------------|-----------------|
| NT-ASGD | 61.2 | 58.8 | 68.7 | 65.6 |
| SWA | 59.1 | 56.7 | 68.1 | 65.0 |
| SWAG | **58.6** | **56.26** | **67.2** | **64.1** |

### 5.5 Regression

Finally, while the empirical focus of our paper is classification calibration, we also compare to additional approximate BNN inference methods which perform well on smaller architectures, including deterministic variational inference (DVI) [47], single-layer deep GPs (DGP) with expectation propagation [4], SGLD [46], and re-parameterization VI [26] on a set of UCI regression tasks. We report test log-likelihoods, RMSEs and test calibration results in Appendix Tables 11 and 12 where it is possible to see that SWAG is competitive with these methods. Additional details are in Appendix 7.

## 6 Discussion

In this paper we developed SWA-Gaussian (SWAG) for approximate Bayesian inference in deep learning. There has been a great desire to apply Bayesian methods in deep learning due to their theoretical properties and past success with small neural networks. We view SWAG as a step towards practical, scalable, and accurate Bayesian deep learning for large modern neural networks.

A key geometric observation in this paper is that the posterior distribution over neural network parameters is close to Gaussian in the subspace spanned by the trajectory of SGD. Our work shows Bayesian model averaging within this subspace can improve predictions over SGD or SWA solutions. Furthermore, Gur-Ari et al. [13] argue that the SGD trajectory lies in the subspace spanned by the eigenvectors of the Hessian corresponding to the top eigenvalues, implying that the SGD trajectory subspace corresponds to directions of rapid change in predictions. In recent work, Izmailov et al. [19] show promising results from directly constructing subspaces for Bayesian inference.

**Acknowledgements**

WM, PI, and AGW were supported by an Amazon Research Award, Facebook Research, NSF IIS-1563887, and NSF IIS-1910266. WM was additionally supported by an NSF Graduate Research Fellowship under Grant No. DGE-1650441. DV was supported by the Russian Science Foundation grant no.19-71-30020. We would like to thank Jacob Gardner and Polina Kirichenko for helpful discussions.

## Footnotes

[2]We ignore momentum for simplicity in this update; however we utilized momentum in the resulting experiments and it is covered theoretically [34].

[3] We note that stochastic gradient Monte Carlo methods [5, 46] also use the SGD trajectory to construct samples from the approximate posterior. However, these methods are principally different from SWAG in that they (1) require adding Gaussian noise to the gradients, (2) decay learning rate to zero and (3) do not construct a closed-form approximation to the posterior distribution, which for instance enables SWAG to draw new samples with minimal overhead. We include comparisons to SGLD [46] in the Appendix.

[4]We use one half as the scale here because both the diagonal and low rank terms include the variance of the weights. We tested several other scales in Appendix 4.

[5] From `sklearn.decomposition.TruncatedSVD`.

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
