[Supplementary Material]

# Supplementary Material: A Simple Baseline for Bayesian Uncertainty in Deep Learning

## 1 Asymptotic Normality of SGD

Under conditions of decaying learning rates, smoothness of gradients, and the existence of a full rank stationary distribution, martingale based analyses of stochastic gradient descent [e.g., 1, Chapter 8] show that SGD has a Gaussian limiting distribution. That is, in the limit as the time step goes to infinity, $t^{1/2}(\theta_t - \theta^*) \to \mathcal{N}(0, \mathcal{H}(\theta)^{-1}\mathbb{E}(\nabla \log p(\theta)\nabla \log p(\theta)^T)\mathcal{H}(\theta)^{-1}))$, where $\mathcal{H}(\theta)^{-1}$ is the inverse of the Hessian matrix of the log-likelihood and $\mathbb{E}(\nabla \log p(\theta)\nabla \log p(\theta)^T)$ is the covariance of the gradients and $\theta^*$ is a stationary point or minima. Note that these analyses date back to Ruppert [23] and Polyak and Juditsky [21] for Polyak-Ruppert averaging, and are still popular in the analysis of stochastic gradient descent.

Mandt et al. [17] and Chen et al. [5] both use the same style of analyses, but for different purposes. We will test the specific assumptions of Mandt et al. [17] in the next section. Finally, note that the technical conditions in these analyses are essentially the same conditions as for the Bernstein von Mises Theorem [e.g., 25, Chapter 10] which implies that the asymptotic posterior will also be Gaussian.

Figure A1: Trajectory of SGD with isotropic Gaussian gradient noise on a quadratic loss function. **Left**: SGD without momentum; **Right**: SGD with momentum.

It may be counter-intuitive that, as we show in Section 4 SWAG captures the geometry of the objective correctly. One might even expect SWAG estimates of variance to be inverted, as gradient descent would oscillate more in the sharp directions of the objective. To gain more intuition about SGD dynamics we visualize SGD trajectory on a quadratic problem. More precisely, we define a 2-dimensional quadratic function $f(x,y) = (x+y)^2 + 0.05 \cdot (x-y)^2$ shown in Figure A1. We then run SGD to minimize this function.

It turns out that the gradient noise plays a crucial role in forming the SGD stationary distribution. If there is no noise in the gradients, we are in the full gradient descent regime, and optimization either converges to the optimizer, or diverges to infinity depending on the learning rate. However, when we add isotropic Gaussian noise to the gradients, SGD converges to the correct Gaussian distribution, as we visualize in the left panel of Figure A1. Furthermore, adding momentum affects the scale of the

distribution, but not its shape, as we show in the right panel of Figure A1. These conclusions hold as long as the learning rate in SGD is not too large.

The results we show in Figure A1 are directly predicted by theory in Mandt et al. [17]. In general, if the gradient noise is not isotropic, the stationary distribution of SGD would be different from the exact posterior distribution. Mandt et al. [17] provide a thorough empirical study of the SGD trajectory for convex problems, such as linear and logistic regression, and show that SGD can often provide a competitive baseline on these problems.

## 1.1 Other Related Work

Given the covariance matrix $A = \mathcal{H}(\theta)^{-1}\mathbb{E}(\nabla \log p(\theta)\nabla \log p(\theta)^T)\mathcal{H}(\theta)^{-1}$, Chen et al. [5] show that a batch means estimator of the iterates (similar to what SWAG uses) themselves will converge to $A$ in the limit of infinite time. We tried batch means based estimators but saw no improvement; however, it could be interesting to explore further in future work.

Intriguingly, the covariance $A$ is the same form as sandwich estimators [see e.g. 19, for a Bayesian analysis in the model mis-specification setting], and so $A = \mathcal{H}(\theta)^{-1}$ under model well-specification [19, 5]. We then tie the covariance matrix of the iterates back to the well known Laplace approximation, which uses $\mathcal{H}(\theta)^{-1}$ as its covariance as described by MacKay [16, Chapter 28], thereby justifying SWAG theoretically as a sample based Laplace approximation.

Finally, in Chapter 4 of Berger [2] constructs an example (Example 10) of fitting a Gaussian approximation from a MCMC chain, arguing that it empirically performs well in Bayesian decision theoretic contexts. Berger [2] give the explanation for this as the Bernstein von Mises Theorem providing that in the limit the posterior will itself converge to a Gaussian. However, we would expect that even in the infinite data limit the posterior of DNNs would converge to something very non-Gaussian - connected modes surrounded by gorges of zero posterior density [8]. One could use this justification to justify fitting a Gaussian from the iterates of SGLD or SGHMC instead.

## 2 Do the assumptions of Mandt et al. [17] hold for DNNs?

In this section, we investigate the results of Mandt et al. [17] in the context of deep learning. Mandt et al. [17] uses the following assumptions:

1. Gradient noise at each point $\theta$ is $\mathcal{N}(0, C)$.

2. $C$ is independent of $\theta$ and full rank.

3. The learning rates, $\eta$, are small enough that we can approximate the dynamics of SGD by a continuous-time dynamic described by the corresponding stochastic differential equation.

4. In the stationary distribution, the loss is approximately quadratic near the optima, i.e. approximately $(\theta - \theta^*)^\top \mathcal{H}(\theta)(\theta - \theta^*)$, where $\mathcal{H}(\theta^*)$ is the Hessian at the optimum; further, the Hessian is assumed to be positive definite.

Assumption 1 is motivated by the central limit theorem, and Assumption 3 is necessary for the analysis in Mandt et al. [17]. Assumptions 2 and 4 may or may not hold for deep neural networks (as well as other models). Under these assumptions, Theorem 1 of Mandt et al. [17] derives the optimal constant learning rate that minimizes the KL-divergence between the SGD stationary distribution and the posterior[1]:

$$\eta^* = 2\frac{B}{N}\frac{d}{tr(C)}, \tag{1}$$

where $N$ is the size of the dataset, $d$ is the dimension of the model, $B$ is the minibatch size and $C$ is the gradient noise covariance.

We computed Equation 1 over the course of training for two neural networks in Figure A.A2a, finding that the predicted optimal learning rate was an order of magnitude larger than what would be used in practice to explore the loss surface in a reasonable time (about $4$ compared to $0.1$).

We now focus on seeing how Assumptions 2 and 4 fail for DNNs; this will give further insight into what portions of the theory do hold, and may give insights into a corrected version of the optimal learning rate.

## 2.1 Assumption 2: Gradient Covariance Noise.

In Figure A.A2b, the trace of the gradient noise covariance and thus the optimal learning rates *are* nearly constant; however, the total variance is much too small to induce effective learning rates, probably due to over-parameterization effects inducing non full rank gradient covariances as was found in Chaudhari and Soatto [4]. We note that this experiment is not sufficient to be fully confident that $C$ is independent of the parameterization near the local optima, but rather that $tr(C)$ is close to constant; further experiments in this vein are necessary to test if the diagonals of $C$ are constant. The result that $tr(C)$ is close to constant suggests that a constant learning rate could be used for sampling in a stationary phase of training. The dimensionality parameter in Equation 1 could be modified to use the number of effective parameters or the rank of the gradient noise to reduce the optimal learning rate to a feasible number.

To estimate $tr(C)$ from the gradient noise we need to divide the estimated variance by the batch size (as $V(\hat{g}(\theta)) = BC(\theta)$), for a correct version of Equation 1. From Assumption 1 and Equation 6 of Mandt et al. [17], we see that

$$\hat{g}(\theta) \approx g(\theta) + \frac{1}{\sqrt{B}}\nabla g(\theta), \nabla g(\theta) \sim N(0, C(\theta)),$$

where $B$ is the batch size. Thus, collecting the variance of $\hat{g}(\theta)$ (the variance of the stochastic gradients) will give estimates that are upscaled by a factor of $B$, leading to a cancellation of the batch size terms:

$$\eta \approx \frac{2}{N}\frac{d}{tr(V(\hat{g}(\theta)))}.$$

To include momentum, we can repeat the analysis in Sections 4.1 and 4.3 of Mandt et al. [17] finding that this also involves scaling the optimal learning rate but by a factor of $\mu$, the momentum term.[2] This gives the final optimal learning rate equation as

$$\eta \approx \frac{2\mu}{N}\frac{d}{tr(V(\hat{g}(\theta)))}. \tag{2}$$

In Figure A2b, we computed $tr(C)$ for VGG-16 and PreResNet-164 on CIFAR-100 beginning from the start of training (referred to as from scratch), as well as the start of the SWAG procedure (referred to in the legend as SWA). We see that $tr(C)$ is never quite constant when trained from scratch, while for a period of constant learning rate near the end of training, referred to as the stationary phase, $tr(C)$ is essentially constant throughout. This discrepancy is likely due to large gradients at the very beginning of training, indicating that the stationary distribution has not been reached yet.

Next, in Figure A2a, we used the computed $tr(C)$ estimate for all four models and Equation 2 to compute the optimal learning rate under the assumptions of Mandt et al. [17], finding that these learning rates are not constant for the estimates beginning at the start of training and that they are too large (1-3 at the minimum compared to a standard learning rate of 0.1 or 0.01).

## 2.2 Assumption 4: Hessian Eigenvalues at the Optima

To test assumption 4, we used a GPU-enabled Lanczos method from GPyTorch [7] and used restarting to compute the minimum eigenvalue of the train loss of a pre-trained PreResNet-164 on CIFAR-100. We found that even at the end of training, the minimum eigenvalue was $-272$ (the maximum eigenvalue was 3580 for comparison), indicating that the Hessian is not positive definite. This result harmonizes with other work analyzing the spectra of the Hessian for DNN training [15, 24]. Further, Garipov et al. [8] and Draxler et al. [6] argue that the loss surfaces of DNNs have directions along which the loss is completely flat, suggesting that the loss is nowhere near a positive-definite quadratic form.

(a) Optimal learning rate.                                      (b) $tr(C)$

Figure A2: Gradient variance norm and computed optimal learning rates for VGG-16 and PreResNet-164. The computed optimal learning rates are always too large by a factor of 10, while the gradient variance stabilizes over the course of training.

# 3 Further Geometric Experiments

In Figure A3 we present plots analogous to those in Section 4 for PreResNet-110 and VGG-16 on CIFAR-10 and CIFAR-100. For all dataset-architecture pairs we see that SWAG is able to capture the geometry of the posterior in the subspace spanned by SGD trajectory.

Figure A3: **Left:** Posterior-density cross-sections along the rays corresponding to different eigenvectors of the SWAG covariance matrix. **Middle:** Posterior-density surface in the plane spanned by eigenvectors of SWAG covariance matrix corresponding to the first and second largest eigenvalues and (**Right:**) the third and fourth largest eigenvalues. Each row in the figure corresponds to an architecture-dataset pair indicated in the title of each panel.

Figure A4: **(a)** 30 samples of SWAG with a rank 20 covariance matches the SWA result over the course of training for PreResNet56 on CIFAR-100. SWAG with a rank of 140, SWAG with a rank of 20, and SWA all outperform ensembles of SGD iterates from the SWA procedure and from a standard SGD training path. **(b)** NLL and **(c)** accuracy by number of samples for WideResNet on CIFAR-100 for SWAG, SWAG-Diag, and SWA. 30 samples is adequate for stable accuracies and NLLs. **(d)** NLL by number of samples for different scales for WideResNet on CIFAR-100 for SWAG, SWAG-Diag, and SWA. Scales beneath 1 perform better, with 0.5 and 0.25 best.

## 4 Hyper-Parameters and Limitations

In this section, we discuss the hyper-parameters in SWAG, as well as some current theoretical limitations.

### 4.1 Rank of Covariance Matrix

We now evaluate the effect of the covariance matrix rank on the SWAG approximation. To do so, we trained a PreResNet56 on CIFAR-100 with SWAG beginning from epoch 161, and evaluated 30 sample Bayesian model averages obtained at different epochs; the accuracy plot from this experiment is shown in Figure A4 (a). The rank of each model after epoch 161 is simply $\min(epoch - 161, 140)$, and 30 samples from even a low rank approximation reach the same predictive accuracy as the SWA model. Interestingly, both SWAG and SWA outperform ensembles of a SGD run and ensembles of the SGD models in the SWA run.

### 4.2 Number of Samples in the Forwards Pass

In most situations where SWAG will be used, no closed form expression for the integral $\int f(y)q(\theta|y)d\theta$, will exist. Thus, Monte Carlo approximations will be used; Monte Carlo integration converges at a rate of $1/\sqrt{K}$, where $K$ is the number of samples used, but practically good results may be found with very few samples (e.g. Chapter 29 of MacKay [16]).

To test how many samples are needed for good predictive accuracy in a Bayesian model averaging task, we used a rank 20 approximation for SWAG and then tested the NLL on the test set as a function of the number of samples for WideResNet28x10 [28] on CIFAR-100.

The results from this experiment are shown in Figure A4 (b, c), where it is possible to see that about 3 samples will match the SWA result for NLL, with about 30 samples necessary for stable accuracy (about the same as SWA for this network). In most of our experiments, we used 30 samples for consistency. In practice, we suggest tuning this number by looking at a validation set as well as the computational resources available and comparing to the free SWA predictions that come with SWAG.

### 4.3 Dependence on Learning Rate

First, we note that the covariance, $\Sigma$, estimated using SWAG, is a function of the learning rate (and momentum) for SGD. While the theoretical work of Mandt et al. [17] suggests that it is possible to optimally set the learning rate, our experiments in Appendix 2 show that currently the assumptions of the theory do not match the empirical reality in deep learning. In practice the learning rate can be chosen to maximize negative log-likelihood on a validation set. In the linear setting as in Mandt et al. [17], the learning rate controls the scale of the asymptotic covariance matrix. If the optimal

learning rate (Equation 1) is used in this setting, the covariance matches the true posterior. To attempt to disassociate the learning rate from the covariance in practice, we rescale the covariance matrix when sampling by a constant factor for a WideResNet on CIFAR-100 shown in Figure A4 (d).

Over several replications, we found that a scale of 0.5 worked best, which is expected because the low rank plus diagonal covariance incorporates the variance twice (once for the diagonal and once from the low rank component).

## 4.4 Necessity of Batch Norm Updates

One possible slowdown of SWAG at inference time is in the usage of updated batch norm parameters. Following Izmailov et al. [11], we found that in order for the averaging and sampling to work well, it was necessary to update the batch norm parameters of networks after sampling a new model. This is shown in Figure A5 for a WideResNet on CIFAR-100 for two independently trained models.

Figure A5: NLL by number of samples for SWAG with and without batch norm updates after sampling. Updating the batch norm parameters after sampling results in a significant improvement in NLL.

## 4.5 Usage in Practice

From our experimental findings, we see that given an equal amount of training time, SWAG typically outperforms other methods for uncertainty calibration. SWAG additionally does not require a validation set like temperature scaling and Platt scaling (e.g. Guo et al. [9], Kuleshov et al. [13]). SWAG also appears to have a distinct advantage over temperature scaling, and other popular alternatives, when the target data are from a different distribution than the training data, as shown by our transfer learning experiments.

Deep ensembles [14] require several times longer training for equal calibration, but often perform somewhat better due to incorporating several independent training runs. Thus SWAG will be particularly valuable when training time is limited, but inference time may not be. One possible application is thus in medical applications when image sizes (for semantic segmentation) are large, but predictions can be parallelized and may not have to be instantaneous.

## 5 Further Classification Uncertainty Results

### 5.1 Reliability Diagrams

We provide the additional reliability diagrams for all methods and datasets in Figure A6. SWAG consistently improves calibration over SWA, and performs on par or better than temperature scaling. In transfer learning temperature scaling fails to achieve good calibration, while SWAG still provides a significant improvement over SWA.

Figure A6: Reliability diagrams (see Section 5.1) for all models and datasets. The dataset and architecture are listed in the title of each panel.

Figure A7: In and out of sample entropy distributions for WideResNet28x10 on CIFAR5 + 5.

Table 1: Symmetrized, discretized KL divergence between the distributions of predictive entropies for data from the first and last five classes of CIFAR-10 for models trained only on the first five classes. The entropy distributions for SWAG are more different than the baseline models.

| Method | JS-Distance |
|---|---|
| SWAG | **3.31** |
| SWAG-Diag | 2.27 |
| MC Dropout | 3.04 |
| SWA | 1.68 |
| SGD (Baseline) | 3.14 |
| SGD + Temp. Scaling | 2.98 |

## 5.2 Out-of-Domain Image Detection

Next, we evaluate the SWAG variants along with the baselines on out-of-domain data detection. To do so we train a WideResNet as described in Section 8 on the data from five classes of the CIFAR-10 dataset, and then analyze their predictions on the full test set. We expect the outputted class probabilities on objects that belong to classes that were not present in the training data to have high-entropy reflecting the model's high uncertainty in its predictions, and considerably lower entropy on the images that are similar to those on which the network was trained.

Table 2: ECE for various versions of SWAG, temperature scaling, and MC Dropout on CIFAR-10 and CIFAR-100.

|  | CIFAR-10 | CIFAR-10 | CIFAR-10 | CIFAR-100 | CIFAR-100 | CIFAR-100 |
|---|---|---|---|---|---|---|
| Model | VGG-16 | PreResNet-164 | WideResNet28x10 | VGG-16 | PreResNet-164 | WideResNet28x10 |
| SGD | $0.0483 \pm 0.0022$ | $0.0255 \pm 0.0009$ | $0.0166 \pm 0.0007$ | $0.1870 \pm 0.0014$ | $0.1012 \pm 0.0009$ | $0.0479 \pm 0.0010$ |
| SWA | $0.0408 \pm 0.0019$ | $0.0203 \pm 0.0010$ | $0.0087 \pm 0.0002$ | $0.1514 \pm 0.0032$ | $0.0700 \pm 0.0056$ | $0.0684 \pm 0.0022$ |
| SWAG-Diag | $0.0267 \pm 0.0025$ | $0.0082 \pm 0.0008$ | $\mathbf{0.0047} \pm 0.0013$ | $0.0819 \pm 0.0021$ | $0.0239 \pm 0.0047$ | $0.0322 \pm 0.0018$ |
| SWAG | $0.0158 \pm 0.0030$ | $\mathbf{0.0053} \pm 0.0004$ | $0.0088 \pm 0.0006$ | $0.0395 \pm 0.0061$ | $0.0587 \pm 0.0048$ | $\mathbf{0.0113} \pm 0.0020$ |
| KFAC-Laplace | $0.0094 \pm 0.0005$ | $0.0092 \pm 0.0018$ | $0.0060 \pm 0.0003$ | $0.0778 \pm 0.0054$ | $\mathbf{0.0158} \pm 0.0014$ | $0.0379 \pm 0.0047$ |
| SWA-Dropout | $0.0284 \pm 0.0036$ | $0.0162 \pm 0.0000$ | $0.0094 \pm 0.0014$ | $0.1108 \pm 0.0181$ | * | $0.0574 \pm 0.0028$ |
| SWA-Temp | $0.0366 \pm 0.0063$ | $0.0172 \pm 0.0010$ | $0.0080 \pm 0.0007$ | $\mathbf{0.0291} \pm 0.0097$ | $0.0175 \pm 0.0037$ | $0.0220 \pm 0.0007$ |
| SGLD | $\mathbf{0.0082} \pm 0.0012$ | $0.0251 \pm 0.0012$ | $0.0192 \pm 0.0007$ | $0.0424 \pm 0.0029$ | $0.0363 \pm 0.0008$ | $0.0296 \pm 0.0008$ |

Table 3: ECE on ImageNet.

| Model | DenseNet-161 | ResNet-152 |
|---|---|---|
| SGD | $0.0545 \pm 0.0000$ | $0.0478 \pm 0.0000$ |
| SWA | $0.0509 \pm 0.0000$ | $0.0605 \pm 0.0000$ |
| SWAG-Diag | $0.0459 \pm 0.0000$ | $0.0566 \pm 0.0000$ |
| SWAG | $0.0204 \pm 0.0000$ | $0.0279 \pm 0.0000$ |
| SWA-Temp | $\mathbf{0.0190} \pm 0.0000$ | $\mathbf{0.0183} \pm 0.0000$ |

To make this comparison quantitative, we computed the symmetrized KL divergence between the binned in and out of sample distributions in Table 1, finding that SWAG and Dropout perform best on this measure. We plot the histograms of predictive entropies on the in-domain (classes that were trained on) and out-of-domain (classes that were not trained on) in Figure A.A7 for a qualitative comparison.

Table 1 shows the computed symmetrized, discretized KL distance between in and out of sample distributions for the CIFAR5 out of sample image detection class. We used the same bins as in Figure A7 to discretize the entropy distributions, then smoothed these bins by a factor of 1e-7 before calculating $KL(\text{IN}||\text{OUT}) + KL(\text{OUT}||\text{IN})$ using the `scipy.stats.entropy` function. We can see even qualitatively that the distributions are more distinct for SWAG and SWAG-Diagonal than for the other methods, particularly temperature scaling.

## 5.3 Tables of ECE, NLL, and Accuracy.

We provide test accuracies (Tables 8,9,10) and negative log-likelihoods (NLL) (Tables 5,6,7) all methods and datasets. We observe that SWAG is competitive with SWA, SWA with temperature scaling and SWA-Dropout in terms of test accuracy, and typically outperforms all the baselines in terms of NLL. SWAG-Diagonal is generally inferior to SWAG for log-likelihood, but outperforms SWA.

In Tables 2,3,4 we additionally report expected calibration error [ECE, 20], a metric of calibration of the predictive uncertainties. To compute ECE for a given model we split the test points into 20 bins based on the confidence of the model, and we compute the absolute value of the difference of the average confidence and accuracy within each bin, and average the obtained values over all bins. Please refer to [20, 9] for more details. We observe that SWAG is competitive with temperature scaling for ECE. Again, SWAG-Diagonal achieves better calibration than SWA, but using the low-rank plus diagonal covariance approximation in SWAG leads to substantially improved performance.

## 6 Language Modeling

We evaluate SWAG using standard Penn Treebank and WikiText-2 benchmark language modeling datasets. Following [18] we use a 3-layer LSTM model with 1150 units in the hidden layer and an embedding of size 400; we apply dropout, weight-tying, activation regularization (AR) and temporal activation regularization (TAR) techniques. We follow [18] for specific hyper-parameter settings such as dropout rates for different types of layers. We train all models for language modeling tasks and evaluate validation and test perplexity. For SWA and SWAG we pre-train the models using standard

Table 4: ECE on CIFAR10 to STL 10.

| Model | VGG-16 | PreResNet-164 | WideResNet28x10 |
|---|---|---|---|
| SGD | $0.2149 \pm 0.0027$ | $0.1758 \pm 0.0000$ | $0.1561 \pm 0.0000$ |
| SWA | $0.2082 \pm 0.0056$ | $0.1739 \pm 0.0000$ | $0.1413 \pm 0.0000$ |
| SWAG-Diag | $0.1719 \pm 0.0075$ | $0.1312 \pm 0.0000$ | $0.1241 \pm 0.0000$ |
| SWAG | $\mathbf{0.1463} \pm 0.0075$ | $\mathbf{0.1110} \pm 0.0000$ | $\mathbf{0.1017} \pm 0.0000$ |
| SWA-Dropout | $0.1803 \pm 0.0024$ | | $0.1421 \pm 0.0000$ |
| SWA-Temp | $0.2089 \pm 0.0055$ | $0.1646 \pm 0.0000$ | $0.1371 \pm 0.0000$ |

Table 5: NLL on CIFAR10 and CIFAR100.

| Dataset | CIFAR-10 | | | CIFAR-100 | | |
|---|---|---|---|---|---|---|
| Model | VGG-16 | PreResNet-164 | WideResNet28x10 | VGG-16 | PreResNet-164 | WideResNet28x10 |
| SGD | $0.3285 \pm 0.0139$ | $0.1814 \pm 0.0025$ | $0.1294 \pm 0.0022$ | $1.7308 \pm 0.0137$ | $0.9465 \pm 0.0191$ | $0.7958 \pm 0.0089$ |
| SWA | $0.2621 \pm 0.0104$ | $0.1450 \pm 0.0042$ | $0.1075 \pm 0.0004$ | $1.2780 \pm 0.0051$ | $0.7370 \pm 0.0265$ | $0.6684 \pm 0.0034$ |
| SWAG-Diag | $0.2200 \pm 0.0078$ | $0.1251 \pm 0.0029$ | $0.1077 \pm 0.0009$ | $1.0163 \pm 0.0032$ | $0.6837 \pm 0.0186$ | $0.6150 \pm 0.0029$ |
| SWAG | $0.2016 \pm 0.0031$ | $\mathbf{0.1232} \pm 0.0022$ | $0.1122 \pm 0.0009$ | $0.9480 \pm 0.0038$ | $\mathbf{0.6595} \pm 0.0019$ | $\mathbf{0.6078} \pm 0.0006$ |
| KFAC-Laplace | $0.2252 \pm 0.0032$ | $0.1471 \pm 0.0012$ | $0.1210 \pm 0.0020$ | $1.1915 \pm 0.0199$ | $0.7881 \pm 0.0025$ | $0.7692 \pm 0.0092$ |
| SWA-Dropout | $0.2328 \pm 0.0049$ | $0.1270 \pm 0.0000$ | $0.1094 \pm 0.0021$ | $1.1872 \pm 0.0524$ | | $0.6500 \pm 0.0049$ |
| SWA-Temp | $0.2481 \pm 0.0245$ | $0.1347 \pm 0.0038$ | $\mathbf{0.1064} \pm 0.0004$ | $1.0386 \pm 0.0126$ | $\mathbf{0.6770} \pm 0.0191$ | $0.6134 \pm 0.0023$ |
| SGLD | $\mathbf{0.2001} \pm 0.0059$ | $0.1418 \pm 0.0005$ | $0.1289 \pm 0.0009$ | $0.9699 \pm 0.0057$ | $0.6981 \pm 0.0052$ | $0.678 \pm 0.0022$ |
| SGD-Ens | $0.1881 \pm 0.002$ | $0.1312 \pm 0.0023$ | $0.1855 \pm 0.0014$ | $\mathbf{0.8979} \pm 0.0065$ | $0.7839 \pm 0.0046$ | $0.7655 \pm 0.0026$ |

Table 6: NLL on ImageNet.

| Model | DenseNet-161 | ResNet-152 |
|---|---|---|
| SGD | $0.9094 \pm 0.0000$ | $0.8716 \pm 0.0000$ |
| SWA | $0.8655 \pm 0.0000$ | $0.8682 \pm 0.0000$ |
| SWAG-Diag | $0.8559 \pm 0.0000$ | $0.8584 \pm 0.0000$ |
| SWAG | $\mathbf{0.8303} \pm 0.0000$ | $\mathbf{0.8205} \pm 0.0000$ |
| SWA-Temp | $0.8359 \pm 0.0000$ | $0.8226 \pm 0.0000$ |

Table 7: NLL when transferring from CIFAR10 to STL10.

| Model | VGG-16 | PreResNet-164 | WideResNet28x10 |
|---|---|---|---|
| SGD | $1.6528 \pm 0.0390$ | $1.4790 \pm 0.0000$ | $1.1308 \pm 0.0000$ |
| SWA | $1.3993 \pm 0.0502$ | $1.3552 \pm 0.0000$ | $1.0047 \pm 0.0000$ |
| SWAG-Diag | $1.2258 \pm 0.0446$ | $1.0700 \pm 0.0000$ | $0.9340 \pm 0.0000$ |
| SWAG | $\mathbf{1.1402} \pm 0.0342$ | $\mathbf{0.9706} \pm 0.0000$ | $\mathbf{0.8710} \pm 0.0000$ |
| SWA-Dropout | $1.3133 \pm 0.0000$ | | $0.9914 \pm 0.0000$ |
| SWA-Temp | $1.4082 \pm 0.0506$ | $1.2228 \pm 0.0000$ | $0.9706 \pm 0.0000$ |

Table 8: Accuracy on CIFAR-10 and CIFAR-100.

| Dataset | CIFAR-10 | | | CIFAR-100 | | |
|---|---|---|---|---|---|---|
| Model | VGG-16 | PreResNet-164 | WideResNet28x10 | VGG-16 | PreResNet-164 | WideResNet28x10 |
| SGD | $93.17 \pm 0.14$ | $95.49 \pm 0.06$ | $96.41 \pm 0.10$ | $73.15 \pm 0.11$ | $78.50 \pm 0.32$ | $80.76 \pm 0.29$ |
| SWA | $93.61 \pm 0.11$ | $96.09 \pm 0.08$ | $\mathbf{96.46} \pm 0.04$ | $74.30 \pm 0.22$ | $\mathbf{80.19} \pm 0.52$ | $82.40 \pm 0.16$ |
| SWAG-Diag | $\mathbf{93.66} \pm 0.15$ | $96.03 \pm 0.10$ | $96.41 \pm 0.05$ | $74.68 \pm 0.22$ | $80.18 \pm 0.50$ | $\mathbf{82.40} \pm 0.09$ |
| SWAG | $93.60 \pm 0.10$ | $96.03 \pm 0.02$ | $96.32 \pm 0.08$ | $\mathbf{74.77} \pm 0.09$ | $79.90 \pm 0.50$ | $82.23 \pm 0.19$ |
| KFAC-Laplace | $92.65 \pm 0.20$ | $95.49 \pm 0.06$ | $96.17 \pm 0.00$ | $72.38 \pm 0.23$ | $78.51 \pm 0.05$ | $80.94 \pm 0.41$ |
| SWA-Dropout | $93.23 \pm 0.36$ | $\mathbf{96.18} \pm 0.00$ | $96.39 \pm 0.09$ | $72.50 \pm 0.54$ | | $82.30 \pm 0.19$ |
| SWA-Temp | $93.61 \pm 0.11$ | $96.09 \pm 0.08$ | $96.46 \pm 0.04$ | $74.30 \pm 0.22$ | $80.19 \pm 0.52$ | $82.40 \pm 0.16$ |
| SGLD | $93.55 \pm 0.15$ | $95.55 \pm 0.04$ | $95.89 \pm 0.02$ | $74.02 \pm 0.30$ | $80.09 \pm 0.05$ | $80.94 \pm 0.17$ |

SGD for 500 epochs, and then run the model for 100 more epochs to estimate the mean $\theta_{\text{SWA}}$ and covariance $\Sigma$ in SWAG. For this experiment we introduce a small change to SWA and SWAG: to estimate the mean $\theta_{\text{SWA}}$ we average weights after each mini-batch of data rather than once per epoch, as we found more frequent averaging to greatly improve performance. After SWAG distribution

Table 9: Accuracy on ImageNet.

| Model | DenseNet-161 | ResNet-152 |
|---|---|---|
| SGD | $77.79 \pm 0.00$ | $78.39 \pm 0.00$ |
| SWA | $\mathbf{78.60} \pm 0.00$ | $78.92 \pm 0.00$ |
| SWAG-Diag | $78.59 \pm 0.00$ | $78.96 \pm 0.00$ |
| SWAG | $78.59 \pm 0.00$ | $\mathbf{79.08} \pm 0.00$ |
| SWA-Temp | $78.60 \pm 0.00$ | $78.92 \pm 0.00$ |

Table 10: Accuracy when transferring from CIFAR-10 to STL-10.

| Model | VGG-16 | PreResNet-164 | WideResNet28x10 |
|---|---|---|---|
| SGD | $\mathbf{72.42} \pm 0.07$ | $75.56 \pm 0.00$ | $76.75 \pm 0.00$ |
| SWA | $71.92 \pm 0.01$ | $\mathbf{76.02} \pm 0.00$ | $\mathbf{77.50} \pm 0.00$ |
| SWAG-Diag | $72.09 \pm 0.04$ | $75.95 \pm 0.00$ | $77.26 \pm 0.00$ |
| SWAG | $72.19 \pm 0.06$ | $75.88 \pm 0.00$ | $77.09 \pm 0.00$ |
| SWA-Dropout | $71.45 \pm 0.11$ | | $76.91 \pm 0.00$ |
| SWA-Temp | $71.92 \pm 0.01$ | $76.02 \pm 0.00$ | $77.50 \pm 0.00$ |

Table 11: Unnormalized test log-likelihoods on small UCI datasets for proposed methods, as well as direct comparisons to the numbers reported in deterministic variational inference (DVI, Wu et al. [27]) and Deep Gaussian Processes with expectation propagation (DGP1-50, Bui et al. [3]), and variational inference (VI) with the re-parameterization trick [12]. * denotes reproduction from [27]. Note that SWAG wins on two of the six datasets, and that SGD serves as a strong baseline throughout.

| dataset | N | D | SGD | SWAG | DVI* | DGP1-50* | VI* | SGLD* | PBP* |
|---|---|---|---|---|---|---|---|---|---|
| boston | 506 | 13 | $-2.536 \pm 0.240$ | $-2.469 \pm 0.183$ | $-2.41 \pm 0.02$ | $\mathbf{-2.33} \pm 0.06$ | $-2.43 \pm 0.03$ | $-2.40 \pm 0.05$ | $-2.57 \pm 0.09$ |
| concrete | 1030 | 8 | $\mathbf{-3.02} \pm 0.126$ | $-3.05 \pm 0.1$ | $-3.06 \pm 0.01$ | $-3.13 \pm 0.03$ | $-3.04 \pm 0.02$ | $-3.08 \pm 0.03$ | $-3.16 \pm 0.02$ |
| energy | 768 | 8 | $-1.736 \pm 1.613$ | $-1.679 \pm 1.488$ | $\mathbf{-1.01} \pm 0.06$ | $-1.32 \pm 0.03$ | $-2.38 \pm 0.02$ | $-2.39 \pm 0.01$ | $-2.04 \pm 0.02$ |
| naval | 11934 | 16 | $6.567 \pm 0.185$ | $\mathbf{6.708} \pm \mathbf{0.105}$ | $6.29 \pm 0.04$ | $3.60 \pm 0.33$ | $5.87 \pm 0.29$ | $3.33 \pm 0.01$ | $3.73 \pm 0.01$ |
| yacht | 308 | 6 | $-0.418 \pm 0.426$ | $\mathbf{-0.404} \pm \mathbf{0.418}$ | $-0.47 \pm 0.03$ | $-1.39 \pm 0.14$ | $-1.68 \pm 0.04$ | $-2.90 \pm 0.01$ | $-1.63 \pm 0.02$ |
| power | 9568 | 4 | $-2.772 \pm 0.04$ | $-2.775 \pm 0.038$ | $-2.80 \pm 0.00$ | $-2.81 \pm 0.01$ | $\mathbf{-2.66} \pm 0.01$ | $-2.67 \pm 0.00$ | $-2.84 \pm 0.01$ |

is constructed we sample and ensemble 30 models from this distribution. We use rank-10 for the low-rank part of the covariance matrix of SWAG distribution.

# 7 Regression

For the small UCI regression datasets, we use the architecture from Wu et al. [27] with one hidden layer with 50 units, training for 50 epochs (starting SWAG at epoch 25) and using 20 repetitions of 90/10 train test splits. We fixed a single seed for tuning before using 20 different seeds for the results in the paper.

We use SGD[3], manually tune learning rate and weight decay, and use batch size of $N/10$ where $N$ is the dataset size. All models predict heteroscedastic uncertainty (i.e. output a variance). In Table 11, we compare subspace inference methods to deterministic VI (DVI, Wu et al. [27]) and deep Gaussian processes with expectation propagation (DGP1-50 Bui et al. [3]). SWAG outperforms DVI and the other methods on three of the six datasets and is competitive on the other three despite its vastly reduced computational time (the same as SGD whereas DVI is known to be 300x slower). Additionally, we note the strong performance of well-tuned SGD as a baseline against the other approximate inference methods, as it consistently performs nearly as well as both SWAG and DVI.

Finally, in Table 11, we compare the calibration (coverage of the 95% credible sets of SWAG and 95% confidence regions of SGD) of both SWAG and SGD. Note that neither is ever too over-confident (far beneath 95% coverage) and that SWAG is considerably better calibrated on four of the six datasets.

Table 12: Calibration on small-scale UCI datasets. Bolded numbers are those closest to 0.95 %the predicted coverage).

|          | N     | D  | SGD               | SWAG              |
|----------|-------|----|-------------------|-------------------|
| boston   | 506   | 13 | $0.913 \pm 0.039$ | $\mathbf{0.936} \pm 0.036$ |
| concrete | 1030  | 8  | $0.909 \pm 0.032$ | $\mathbf{0.930} \pm 0.023$ |
| energy   | 768   | 8  | $0.947 \pm 0.026$ | $\mathbf{0.951} \pm 0.027$ |
| naval    | 11934 | 16 | $\mathbf{0.948} \pm 0.051$ | $0.967 \pm 0.008$ |
| yacht    | 308   | 6  | $0.895 \pm 0.069$ | $\mathbf{0.898} \pm 0.067$ |
| power    | 9568  | 4  | $\mathbf{0.956} \pm 0.006$ | $0.957 \pm 0.005$ |

# 8    Classification Experimental Details and Parameters

In this section we describe all of the architectures and hyper-parameters we use in Sections 5.1, 5.2.

On ImageNet we use architecture implementations and pre-trained weights from `https://github.com/pytorch/vision/tree/master/torchvision`. For the experiments on CIFAR datasets we adapted the following implementations:

- VGG-16:    `https://github.com/pytorch/vision/blob/master/torchvision/models/vgg.py`

- Preactivation-ResNet-164:    `https://github.com/bearpaw/pytorch-classification/blob/master/models/cifar/preresnet.py`

- WideResNet28x10:    `https://github.com/meliketoy/wide-resnet.pytorch/blob/master/networks/wide_resnet.py`

For all datasets and architectures we use the same piecewise constant learning rate schedule and weight decay as in Izmailov et al. [11], except we train Pre-ResNet for 300 epochs and start averaging after epoch 160 in SWAG and SWA. For all of the methods we are using our own implementations in PyTorch. We describe the hyper-parameters for all experiments for each model:

**SWA**    We use the same hyper-parameters as Izmailov et al. [11] on CIFAR datasets. On ImageNet we used a constant learning rate of $10^{-3}$ instead of the cyclical schedule, and averaged 4 models per epoch. We adapt the code from `https://github.com/timgaripov/swa` for our implementation of SWA.

**SWAG**    In all experiments we use rank $K = 20$ and use 30 weight samples for Bayesian model averaging. We re-use all the other hyper-parameters from SWA.

**KFAC-Laplace**    For our implementation we adapt the code for KFAC Fisher approximation from `https://github.com/Thrandis/EKFAC-pytorch` and implement our own code for sampling. Following [22] we tune the scale of the approximation on validation set for every model and dataset.

**MC-Dropout**    In order to implement MC-dropout we add dropout layers before each weight layer and sample 30 different dropout masks for Bayesian model averaging at inference time. To choose the dropout rate, we ran the models with dropout rates in the set $\{0.1, 0.05, 0.01\}$ and chose the one that performed best on validation data. For both VGG-16 and WideResNet28x10 we found that dropout rate of $0.05$ worked best and used it in all experiments. On PreResNet-164 we couldn't achieve reasonable performance with any of the three dropout rates, which has been reported from the work of He et al. [10]. We report the results for MC-Dropout in combination with both SWA (SWA-Drop) and SGD (SGD-Drop) training.

**Temperature Scaling**    For SWA and SGD solutions we picked the optimal temperature by minimizing negative log-likelihood on validation data, adapting the code from `https://github.com/gpleiss/temperature_scaling`.

**SGLD**  We initialize SGLD from checkpoints pre-trained with SGD. We run SGLD for 100 epochs on WideResNet and for 150 epochs on PreResNet-156. We use the learning rate schedule of [26]:

$$\eta_t = \frac{\eta_0}{(\eta_1 + t)^{0.55}}.$$

We tune constants $a, b$ on validation. For WideResNet we use $a = 38.0348$, $b = 13928.7$ and for PreResNet we use $a = 40.304$, $b = 15476.4$; these values are selected so that the initial learning rate is 0.2 and final learning rate is 0.1. We also had to rescale the noise in the gradients by a factor of $5 \cdot 10^{-4}$ compared to [26]. Without this rescaling we found that even with learning rates on the scale of $10^{-7}$ SGD diverged. We note that noise rescaling is commonly used with stochastic gradient MCMC methods (see e.g. the implementation of [29]).

On CIFAR datasets for tuning hyper-parameters we used the last 5000 training data points as a validation set. On ImageNet we used 5000 of test data points for validation. On the transfer task for CIFAR10 to STL10, we report accuracy on all 10 STL10 classes even though frogs are not a part of the STL10 test set (and monkeys are not a part of the CIFAR10 training set).

## Footnotes

[1]An optimal diagonal preconditioner is also derived; our empirical work applies to that setting as well. A similar analysis with momentum holds as well, adding in only the momentum coefficient.

[2]Our experiments used $\mu = 0.1$ corresponding to $\rho = 0.9$ in PyTorch's SGD implementation.

[3]Except for concrete where we use Adam due to convergence issues.