[Reviews · NeurIPS 2019]

Reviewer 1



The method is almost trivially simple, scalable and easy to implement, yet the empirical evaluation shows that it performs competitively and often better than all alternatives. This is the best kind of paper! The task of representing uncertainty over model weights is highly significant -- it is debatably *the* core problem in Bayesian deep learning, with (as the authors point out) applications to calibrated decision making, out-of-sample detection, adversarial robustness, transfer learning, and more. I expect this baseline to be widely used by researchers in the field, and likely implemented by practitioners as well. The paper is well written and easy to follow. The method is clearly motivated and cleanly presented. The experimental results are extensive and compelling, and include comparisons to the major alternative approaches from recent literature. I appreciate that the experiments include some 'real' tasks (e.g., Imagenet models) as opposed to the toy problems often used in Bayesian deep learning papers. One omission I found fairly glaring was the lack of any discussion of the seemingly even simpler baseline of iterate ensembles. If you've got a bunch of SGD iterates lying around, why bother fitting a Gaussian and ensembling its samples, when you could have just ensembled the iterates directly? I'd expect that imposing the Gaussian distributional assumption increases bias and reduces variance, and I could be convinced that in practical situations the bias-variance tradeoff is always in favor of fitting the Gaussian, but I want to see that comparison! Since the main innovation of the paper is to fit a Gaussian to SGD iterates, the question of whether doing so *actually helps* seems quite foundational. Why not include iterate ensembling as a baseline in Figs 2 and 3? Update: thanks to the authors for your response and the new results, which are encouraging. I'm still not sure I have good intuition for when (and why) SWAG will outperform iterate ensembles -- I would appreciate some discussion on this point in the paper -- but it's good to see evidence that it often does. I also appreciated Reviewer 3's points re convergence to an isotropic Gaussian -- having not read Mandt et al. closely, I didn't realize that under non-crazy assumptions (i.e., that the data are generated from the model) the scale of the SGD iterate distribution is independent of the shape of the true posterior. Just reading this submission (e.g. section 3), it's easy to believe otherwise; readers would be better served if the paper clarified the gap between the theory and the empirical results. R3 also makes a very strong point that the paper should discuss how the SWAG learning rates are chosen, since Mandt et al. (in the settings of both eqn (13) and section 6.2) indicates that the learning rate is crucial to the scale of posterior uncertainty. Overall I still favor accepting the paper. Whether or not we consider SWAG a principled Bayesian approximation, it's significant that a simple method performs well at the tasks used to benchmark Bayesian deep learning algorithms, and asking new methods to do better is a reasonable challenge. To the extent that SWAG is a broken approximation, it should be possible to beat it, but in the meantime simple baselines are important. I think this paper will help move the field forward. That said, if it's accepted I encourage the authors to consider softening the Bayesian framing; positioning SWAG as 'merely' a useful approach to quantifying uncertainty seems like a much stronger case.

Reviewer 2



The authors propose a very simple and practical method for evaluating Bayesian model uncertainty by running SGD with a fixed step-size and fitting a low-rank Gaussian to the resulting iterates. Surprisingly, this method works better than many more involved methods for approximating Bayesian inference. This method is currently one of the most practical ways of evaluating model/parameter uncertainty when training deep neural networks and could be very useful to many people. The paper is written very clearly. There is useful theory supporting the method, with relevant references to Mandt et al about the approximately Gaussian distribution of SGD and how this relates to the Bayesian posterior. There is a useful experimental evaluation of the posterior landscape. The method is shown to work at Imagenet scale. There is a thorough comparison against competing methods like Laplace approximations, SGLD and dropout.

Reviewer 3



The paper is well written, and the empirical evaluation is thorough. However I am concerned by the interpretation the authors propose, which I think will add confusion to the literature. I feel that the paper should not be accepted without significant re-structuring. The authors describe their method as a baseline for Bayesian uncertainty, however it is straightforward to show in simple cases that the stationary distribution of SGD does not approximate the Bayesian posterior. Indeed Mandt et al. ('SGD as approximate Bayesian inference') showed that, under Bernstein-von Mises type identifiability assumptions, SGD will converge to an isotropic Gaussian distribution near the minimum as the dataset size grows. This Gaussian distribution is independent of the Hessian of the posterior, depending only on the batch size and the learning rate. In order to get Bayesian samples from SGD, one must replace the learning rate by a learned pre-conditioning matrix. On a more minor note, I would have appreciated a comparison between SWAG and conventional model ensembling. While ensembling requires k times more computation at training time, it requires the same computation as SWAG at test time, and I would expect it to perform substantially better on the test set. Edit: I thank the authors for their response, however I remained concerned about the framing of the paper as a Bayesian method. In my view, it is not appropriate to call any distribution over parameters Bayesian. One should show that the distribution is close to a valid posterior under some reasonable (if often unrealistic) assumptions. By contrast, it is possible to prove that under reasonable assumptions, the stationary distribution of SGD is independent of the shape of the posterior. Furthermore, the authors themselves demonstrate that Mandt et al's method for predicting the learning rate fails in practice. I presume that this learning rate, which is what sets the uncertainty scale, must be tuned on the validation set, although the authors claim otherwise in the paper. I do agree with the other reviewers that the proposed method provides a strong baseline for handling uncertainty in deep learning. However I believe that the paper should be re-worked for a future conference, emphasising that the method is primarily empirical, rather than being a Bayesian approximation. I am particularly concerned that this paper furthers existing misunderstandings about the Mandt paper in the community. With this in mind, the authors should compare against other non-Bayesian baselines for uncertainty. To my knowledge, the "deep ensembles" method already outperforms all the Bayesian techniques considered in this work, and additionally I remain unconvinced that SWAG would outperform ensembling in settings where the bottleneck is the compute cost at test time. Finally, I note that SGLD was run at a reduced temperature below 1 in order to prevent the iterates diverging. This may indicate that the posterior is improper, in which case any accurate Bayesian method would be likely to produce poor results.

[Author Response · NeurIPS 2019]

We would like to thank all the reviewers for thoughtful feedback. As the reviewers pointed out, SWAG is a very practical
Bayesian deep learning method readily applicable to ImageNet-scale problems. SWAG achieves strong results on image
classification, tabular regression and language modeling, out-performing strong and elaborate Bayesian deep learning
methods. We also explicitly demonstrate that SWAG can capture the shape of the posterior (along certain directions) in
Section 4, which justifies using SWAG as an approximation to the posterior distribution. We believe that our paper
(i) sets a strong baseline for Bayesian deep learning and (ii) motivates researchers in the field to conduct realistic
evaluations on large-scale datasets and models, and (iii) use loss surface visualizations to show that the approximate
posterior distribution captures the shape of the true posterior.

Inspired by reviewer suggestions, we ran two additional experiments. First, we evaluated **ensembles of SGD iterates**
that were used to construct the SWAG approximation for all of our CIFAR models. We report NLLs in the table:

|  | CIFAR-100 | | CIFAR-10 | |
|---|---|---|---|---|
| Architecture | SWAG | SGD-Ens | SWAG | SGD-Ens |
| VGG-16 | $0.9480 \pm 0.0038$ | $\mathbf{0.8979} \pm 0.0065$ | $0.2016 \pm 0.0031$ | $\mathbf{0.1883} \pm 0.002$ |
| PreResNet-164 | $\mathbf{0.6595} \pm 0.0019$ | $0.7839 \pm 0.0046$ | $\mathbf{0.1232} \pm 0.0022$ | $0.1312 \pm 0.0023$ |
| WideResNet28x10 | $\mathbf{0.6078} \pm 0.0006$ | $0.7655 \pm 0.0026$ | $\mathbf{0.1122} \pm 0.0009$ | $0.1855 \pm 0.0014$ |

SWAG loses on VGG-16, but wins by a large margin on the larger PreResNet-164 and WideResNet28x10; the results for
accuracy and ECE are analogous. We will include these results as well as results on ImageNet and transfer learning in the
camera-ready version. Second, we evaluated **ensembles of independently trained SGD solutions** on PreResNet-164
on CIFAR-100. We found that an ensemble of 3 SGD solutions has high accuracy ($82.1\%$), but only achieves NLL
$0.6922$, which is *worse than a single SWAG solution*. An ensemble of 5 SGD solutions achieves NLL $0.6478$, which
is *competitive with a single SWAG solution, that requires* $5\times$ *less computation to train*. Moreover, we can similarly
ensemble independently trained SWAG models; an ensemble of 3 SWAG models achieves NLL of $0.6178$.

**R1**: We thank the reviewer for the thoughtful and positive review. In addition to the new results we discuss above, we
note that in appendix Figure 3a we show that in terms of accuracy SWAG outperforms an ensemble of SGD iterates.
We would also like to note that in many problems, such as incremental learning (see e.g. [1]), it is desirable to represent
uncertainty over weights as a closed form distribution, rather than just storing samples. Further, we can produce an
arbitrary number of samples from a fixed SWAG approximation, and in appendix Figure 3b, we show that NLL of the
ensemble continues to improve as we add more samples. With just using ensembles of SGD iterates, we cannot cheaply
increase the ensemble.

**R2**: Thank you for the thoughtful and positive review. See the above comparison with SGD-ensembles. [2] demonstrated
that high-frequency ensembles of SGD iterates typically outperform snapshot ensembles, so we focus on the former.

**R3**: While we value the feedback, and are happy you appreciate the quality of the work, we do not agree that the
paper should be rejected unless SWAG is not called "an approximation to Bayesian learning". The proposed method
is unequivocally an approximate Bayesian inference approach, exactly analogous to the Laplace approximation or
variational methods. Similar to many such canonical approximate Bayesian inference procedures, we use a Gaussian
approximation to the posterior, but centred on the SWA solution, with curvature defined by the SGD trajectory; for
comparison, the Laplace approximation uses a Gaussian centred on the SGD solution with curvature defined by the
Hessian of the posterior log-density at that point. Whether or not the posterior is truly Gaussian (as modeled by
Laplace or SWAG), or whether the Gaussian should be centred at an SGD solution (as in Laplace), or what its curvature
should be, or whether the stationary distribution of SGD is Gaussian, are reasonable questions for Laplace, variational
approaches, SWAG, and many other methods, but orthogonal to whether these methods provide approximate Bayesian
inference. It is fair to question the assumptions – indeed we do so ourselves in the paper, and provide exhaustive
empirical support in Section 4 – but calling SWAG an approximate Bayesian method is factually correct and thus not a
fair reason for rejection. Moreover, the assumptions of our procedure are much milder than many standard approximate
Bayesian inference procedures, such as the widespread mean-field variational approximations which assume fully
factorized posteriors. While, as you mention, it is possible to construct special cases where the stationary distribution
does not capture the shape of the posterior (Section 6.2 of [3]), in general these distributions are tightly constrained as
in equation (13) of [3]. In Section 4 of the paper (in particular Figures 1 and Appendix Figure 2) we go beyond many
works employing Gaussian posterior approximations to explicitly demonstrate that the posterior for our applications is
approximately Gaussian in the PCA subspace of the SGD trajectory and SWAG is able to capture its shape.

We evaluated ensembles of independent SGD solutions as you suggested; please see the discussion above.

**References**:

[1] Kirkpatrick, James, et al. Overcoming catastrophic forgetting in neural networks. PNAS, 2017.
[2] Garipov, Timur, et al. Loss Surfaces, Mode Connectivity, and Fast Ensembling of DNNs. NeurIPS, 2018.
[3] Mandt, Stephan, et al. Stochastic Gradient Descent as Approximate Bayesian Inference. JMLR, 2017.


[Meta-Review · NeurIPS 2019]

This paper presents SWAG, a method that uses the iterates of a Polyak-averaging-like stochastic gradient descent to approximate the posterior distribution of a neural network. It is presented as a simple baseline for uncertainty in large deep neural networks and the authors demonstrate its effectiveness on a variety of large scale tasks including residual networks on CIFAR and Imagenet. The strengths of this paper are: - it is indeed a simple baseline for a promising area of research that is really lacking good baselines - experiments are thorough and on benchmarks that are large and interesting to the wider deep learning community - the authors empirically evaluate the quality of their approximation and provide some analysis The main criticism of this paper is that it is not really Bayesian from a purist perspective. R3 is correct to point out that the presented approximation can not actually capture the true posterior as shown by Mandt et al. (Stochastic Gradient Descent as Approximate Bayesian Inference). The language of the paper at times implies otherwise and R3 is right to point this out (e.g. L192 "our procedure... corresponds to fully Bayesian inference"). It also is rather close to Mandt et al. in methodology. The major difference appears to be the application to deep neural networks, the scale of which justifies the approximations presented here. The author's treatment of Mandt et al. in related work is not entirely fair and R3 is right to point this out. That paper explores iterate averaging and Algorithm 1 details a version that doesn't involve a full-covariance matrix. The reason the authors use a full covariance later in the paper is because they show mathematically that one cannot capture the posterior using SGD iterates without doing so. The recommendation is accept, because the empirical work is thorough, this area does indeed lack reasonable baselines and the authors demonstrate empirically that their method gives a reasonable approximation. However, we request that the authors make clear that this is an approximation and especially please give proper attribution to Mandt et al. in the camera ready version.